# Probabilistic Temporal Sampling for Anomaly Detection in Ethereum Networks

## Abstract

The rapid growth of the Ethereum network necessitates advanced anomaly detection techniques to enhance security, transparency, and resilience against evolving malicious activities. While there have been significant strides in anomaly detection, they often fall short in capturing the intricate spatial-temporal patterns inherent in blockchain transactional data. This study presents a scalable framework that integrates Graph Convolutional Networks (GCNs) with Temporal Random Walks (TRW) specifically designed to adapt to the complexities and temporal dynamics of the Ethereum transaction network. Unlike traditional methods that focus on detecting specific attack types, such as front-running or flash loan exploits, our approach targets time-sensitive anomalies more broadly—detecting irregularities such as rapid transaction bursts, anomalous token swaps, and sudden volume spikes. This broader focus reduces reliance on pre-defined attack categories, making the method more adaptable to emerging and evolving malicious strategies. To ground our contributions, we establish three theoretical results: (1) the effectiveness of TRW in enhancing GCN-based anomaly detection by capturing temporal dependencies, (2) the identification of weight cancellation conditions in the anomaly detection process, and (3) the scalability and efficiency improvements of GCNs achieved through probabilistic sampling. Empirical evaluations demonstrate that the TRW-GCN framework outperforms state-of-the-art Temporal Graph Attention Networks (TGAT) in detecting time-sensitive anomalies. Furthermore, as part of our ablation study, we evaluated various anomaly detection techniques on the TRW-GCN embeddings and found that our proposed scoring classifier consistently achieves higher accuracy and precision compared to baseline methods such as Isolation Forest, One-Class SVM, and DBSCAN, thereby validating the robustness and adaptability of our framework.

## 1 Introduction

The Ethereum network is a dynamic and complex ecosystem, characterized by high-frequency transactions, time-sensitive interactions, and evolving patterns of fraudulent activity. Anomalous behaviors such as flash loans, front-running attacks, and MEV (Miner Extractable Value) bots pose significant threats to the security and integrity of the network. These behaviors often unfold over time, making it essential to account for temporal correlations in transaction patterns for effective anomaly detection. The dynamic nature of Ethereum presents unique challenges that cannot be fully addressed using static graph analysis or traditional machine learning approaches.

Graph Convolutional Networks (GCNs) have emerged as a transformative tool in the domain of graph-structured data representation. Their ability to encapsulate both local and global graph structures has paved the way for their application in diverse fields. While traditional GCNs have shown remarkable potential in handling static graph structures, their application to dynamic graphs introduces new

challenges and opportunities. In order to extend GCNs to dynamic graphs, it is crucial to understand how learning on dynamic graphs works, which is a relatively recent area of research. There have been studies which investigate discrete-time graphs represented as a sequence of graph snapshots (1) (2) (3). Also several continuous-time approaches have been presented (4) (5) (6) (7) (8), where continous dynamic graphs means that edges can appear at any time (8) (9).

Also, the topic of anomaly detection in Blockchain has received considerable attention. For example, in Ethereum, the unexpected appearance of particular subgraphs has implied new malware (10). Anomaly detection in blockchain transaction networks is an emerging area of research in the cryptocurrency community (11). Wu et al. (12) investigated phishing detection in blockchain network using unsupervised learning algorithms. Ofori-Boateng et al. (13) have also discussed topological anomaly detection in multilayer blockchain networks. Given that the Ethereum network witnesses dynamically evolving transaction patterns, it becomes imperative to account for the temporal sequences and correlations of transactions. Unlike general-purpose graph neural networks, TRW-GCN is a domain-specific framework tailored to the Ethereum network's unique dynamics, see Table 1 for comparison. By leveraging temporal features and dynamic embeddings, our approach enables the detection of time-sensitive anomalies such as flash loans and MEV bots, with minimal computational complexity. Our research offers several contributions:

**Enhanced Anomaly Detection Effectiveness**: Our model leverages TRW in tandem with GCN to improve anomaly detection effectiveness. This integration improves the detection of anomalies in the Ethereum transaction network by effectively leveraging temporal information embedded within transaction patterns. The model's ability to analyze temporal correlations allows it to identify anomalies that traditional methods often overlook.

**Efficiency in Sampling Representative Nodes**: Given the substantial size and continuous growth of the Ethereum blockchain, efficient sampling methods are essential. Our TRW-GCN provides a solution that balances accuracy with computational efficiency. Many temporal graph learning frameworks face performance bottlenecks when applied to densely connected graphs; for instance, models such as TGAT (4) and AddGraph (14) incorporate temporal dynamics but often come with high computational costs and are sensitive to the quality of temporal features, which can limit their applicability to Ethereum's specific requirements, whereas TRW-GCN prioritize edges based on their timestamps, enabling the model to capture time-sensitive relationships without the overhead of attention mechanisms used in models like TGAT. See Table 1 for comparison.

**Detecting Patterns Leading to Sophisticated Attacks**: While existing works like "Flash Boys 2.0" (15) and "Combatting Front-Running in Smart Contracts" (16) which focus on detecting front-running attacks specifically, our approach targets time-sensitive anomalies more broadly. These anomalies include behaviors that may precede or indirectly relate to specific exploits, such as Front-Running Transactions, Flash Loan Exploits, High-Frequency Token Swaps, and Irregular Contract Interactions, see Table 5 for definitions. By identifying these timing-dependent irregularities, our work addresses a wider range of anomalous behaviors that are indicative of potential security threats.

Table 1: Comparison of TRW-GCN with Existing Temporal GNNs in Blockchain context

| Aspect | TGAT | AddGraph | TRW-GCN |
|---|---|---|---|
| Temporal Modeling Mechanism | Temporal attention on time-encoded node embeddings | Time-decay functions over temporal edges | Temporal random walks to construct time-aware neighborhoods |
| Domain Specialization (Ethereum) | General-purpose model with time-aware positional encodings and attention mechanisms | General-purpose, may underperform TGAT, Less interpretable than attention models | Tailored for Ethereum with attention to domain-specific phenomena (e.g., flash loans, MEV, wash trading) |
| Anomaly Type Detection | Detects broad irregularities, limited granularity, Heavy computation due to multi-head | Captures gradual shifts, not sharp transaction bursts | Detects fine-grained, time-sensitive anomalies like front-running and high-frequency exploits |
| Robustness to Transaction Bursts | Limited; signal may be diluted by attention weights | Time-decay may smooth over bursts | High; TRW preserves burst patterns in short temporal windows |
| Real-World Applicability to Ethereum | Rare in blockchain studies; lacks deployment cases | Not used in Ethereum networks | Demonstrates superior results in transaction-based anomaly detection |

## 2   Model Design

GCNs are a pivotal neural network architecture crafted specifically for graph-structured data. Through the use of graph convolutional layers, we seamlessly aggregate information from neighboring nodes and edges to refine node embeddings. In enhancing this mechanism, we incorporate probabilistic

sampling, which proves particularly adept in analyzing the vast Ethereum network. The incorporation of TRW adds a rich layer to this framework. TRW captures the temporal sequences in Ethereum transactions and not only focuses on nodes' spatial prominence but also considers the transactional chronology. Here, 'time' is conceptualized based on the sequence and timestamps of Ethereum transactions, leading to a dynamically evolving, time-sensitive representation of the network.

Here, graph is represented as $G = (V, E)$, where $V$ is the set of nodes (vertices) and E is the set of edges connecting the nodes. Each node $v_i$ in the graph is associated with a feature vector $F_i$, and $F \in \mathbb{R}^{|V| \times 4}$ represents a feature matrix of size 4. Aggregation is a process to combine the feature vectors of neighboring nodes using an adjacency matrix $A$ to capture graph connectivity. To enable information propagation across multiple layers, the graph convolution operation is performed iteratively through multiple graph convolutional layers (GCLs). The output of one layer serves as the input to the next layer, allowing the propagation of information through the network. The node representations are updated layer by layer, allowing information from neighbors and their neighbors to be incorporated into the node features. The parameters $W^l$ are learned during the training process to optimize the model's performance on a specific graph-based task. GCNs often consist of multiple layers, where each layer iteratively updates the node representations:

$$h_i^{(l)} = Activation \left( W^{(l)} \text{Aggregate} \left( h_j^{(l-1)} | j \in N(i) \right) \right) \tag{1}$$

Here, $h_i^{(l)}$ is the representation of node $i$ at layer $l$, and $h_j^{(l-1)}$ is the representation of neighboring node $j$ at the previous layer $(l-1)$. The final layer is usually followed by a global pooling operation to obtain the graph-level representation. The pooled representation is then used to make predictions.

## 2.1 Incorporating TRW into GCN

The TRW-enhanced GCN creates a multidimensional representation that captures both the structural intricacies and time-evolving patterns of transactions. Such an approach requires meticulous mathematical modeling to substantiate its efficacy, and exploring the depths of this amalgamation can reveal further insights into the temporal rhythms of the Ethereum network.

**Temporal Random Walk (TRW)**

Given a node $i$, the probability $P_{ij}$ of moving to a neighboring node $j$ can be represented as:

$$P_{ij} = \frac{\omega_{ij}}{\sum_k \omega_{ik}} \tag{2}$$

where $\omega_{ij}$ is the weight of the edge between node $i$ and $j$, and the denominator is the sum of weights of all edges from node $i$. In a TRW, transition probabilities take into account temporal factors. Let's define the temporal transition matrix $T$ where each entry $T_{ij}$ indicates the transition probability from node $i$ to node $j$ based on temporal factors.

$$T_{ij} = \alpha \times A_{ij} + (1 - \alpha) \times f(t_{ij}) \tag{3}$$

where $A_{ij}$ is the original adjacency matrix's entry for nodes $i$ and $j$. $\alpha$ is a weighting parameter. $f_{ij}$ is a function of the temporal difference between node $i$ and node $j$. The temporal weighting function could be defined as an exponentially decaying function:

$$f(t_{ij}) = exp(-\gamma \cdot t_{ij}) \tag{4}$$

where $\gamma > 0$ is a decay hyperparameter that controls how sensitive the model is to temporal differences. $f(t_{ij}) \in [0, 1]$, with values closer to 1 for temporally close nodes and closer to 0 for nodes that are far apart in time. Given this temporal transition matrix $T$, a normalized form $\widetilde{T}$ can be used for a GCN layer:

$$\widetilde{T} = \widetilde{D}_T^{-1} T \tag{5}$$

where $\widetilde{D}_T$ is the diagonal degree matrix of $T$. To incorporate the TRW's temporal information into the GCN, we can modify the original GCN operation using $\widetilde{T}$ :

$$h^{(l+1)} = \sigma \left( \widetilde{D}_T^{-\frac{1}{2}} \widetilde{T} \widetilde{D}_T^{-\frac{1}{2}} h^{(l)} W^{(l)} \right) \tag{6}$$

## 2.2 Effect on Anomaly Detection

The embeddings from a GCN (post TRW influence) should be more sensitive to recent behaviors and patterns. When these embeddings are passed to a classifier, clustering and scoring algorithms like DBSCAN, OCSVM, ISOLATION FOREST, and LOF, anomalies that are based on recent or time-sensitive behaviors are more likely to stand out. In our work, the term "anomaly" refers to patterns that are statistically uncommon or divergent from the norm based on the features learned by our model. These uncommon patterns, while not definitively erroneous, are of interest because they deviate from typical behavior. In the context of Ethereum transactions, such deviations indicate suspicious activities, novel transaction patterns, or transaction bursts.

While we here provide insight and mathematical proofs, the true value of TRW in improving GCN over traditional sampling is empirical. We will compare the performance of GCN with and without TRW on a temporal dataset to see tangible benefits (see appendix B.5). Here is how temporal weights are applied:

1. Node Features are weighted by time: When updating the node features through the matrix multiplication, nodes that are temporally closer influence each other more, allowing recent patterns to be highlighted.

2. Temporal Relationships are captured: The modified node features inherently capture temporal relationships because they aggregate features from temporally relevant neighbors.

3. Higher Sensitivity to recent anomalies: With temporal weighting, anomalies that have occurred recently will be more pronounced in the node feature space.

**Theorem 1:** Let $G = (V, E)$ be a graph with node features $h_i \in \mathbb{R}^d$ for $i \in V$, and let a GCN generate embeddings through neighbor aggregation. Incorporating TRW, represented by a temporal weight matrix $T$, into the aggregation mechanism enhances the effectiveness of detecting temporally influenced anomalies. Specifically, if $T$ encodes temporal transitions such that $T_{ij} \neq 1$ for all $i, j$, the feature representation $h_i^{(l+1)}$ for an anomalous node $n$ differs significantly from the non-temporal case:

$$\|h'_n - h_n\|_2 > \delta, \tag{7}$$

for some sensitivity threshold $\delta > 0$, where $h_n$ is the embedding without TRW and $h'_n$ is the embedding with TRW.

**Proof.**

Anomaly detection is the task of distinguishing outliers from normal data points in a given feature space. If we have an anomaly score function $s : \mathbb{R}^d \to \mathbb{R}$, we can detect anomalies by: $s(v) > \delta$ Where $\delta$ is a threshold, and $\underline{v}$ is a vector in the feature space.

A GCN produces node embeddings (or features) by aggregating information from a node's neighbors in the graph. Let's express this aggregation for a single node using a simple form of a GCN layer:

$$h_i^{(l+1)} = \sigma \left( \sum_{j \in Neighbors(i)} W h_j^{(l)} \right) \tag{8}$$

Where $\underline{h_i}^{(l)}$ is the feature of node $\underline{i}$ at layer $\underline{l}$, and $\underline{W}$ is the weight matrix.

**Incorporating TRW:** With a temporal random walk, the aggregation process is influenced by time, so the aggregation becomes:

$$h_i^{(l+1)} = \sigma \left( \sum_{j \in Neighbors(i)} T_{ij} W h_j^{(l)} \right) \tag{9}$$

Where $\underline{T_{ij}}$ is the temporal transition probability from node $\underline{j}$ to node $\underline{i}$. Let's assume a node with an anomaly will have a different feature vector from the nodes without anomalies. For simplicity, let's use the Euclidean distance as the anomaly score: $s(v) = \|v - \mu\|$ where $\mu$ is the mean vector of all node features. Given a temporal anomaly (an anomaly that's influenced by recent events), using TRW will result in a modified feature vector for the anomalous node. Let's consider two scenarios:

1. GCN without TRW: For an anomalous node $\underline{n}$, its feature vector is: $h_n = \sigma \left( \sum_j W h_j \right)$

2. GCN with TRW: For the same anomalous node $\underline{n}$, it becomes: $h'_n = \sigma \left( \sum_j T_{nj} W h_j \right)$

If the anomaly is temporally influenced, then $h'_n$ should be significantly different from $h_n$ due to the weights introduced by $\underline{T_{nj}}$ (see Theorem 2 for weight cancellation). In the context of our anomaly score function: $s(h'_n) - s(h_n) > \delta$ where $\delta$ is a value indicating the sensitivity of the temporal context; we will use this later in our scoring method. If the anomaly is truly temporally influenced, this difference will be significant, and thus, the GCN with TRW will have a higher likelihood of detecting the anomaly. From the linear algebra perspective, the effect of TRW on a GCN for anomaly detection is evident in how node features are aggregated. The temporal weights (from $T_{ij}$) make the GCN more sensitive to temporal influences, making it more adept at detecting anomalies. The theoretical result in Theorem 1 holds for any d-dimensional feature vector, including the 10-dimensional vectors used in the empirical section.

**Theorem 2:** Let $\mathbb{R}^m$ be a vector space, and let $h_n \in \mathbb{R}^m$ represent a feature vector. Define a temporal transformation matrix $T_{nj} \in \mathbb{R}^{m \times m}$, where each entry $t_{ij}$ encodes the temporal weights. Let $h'_n = T_{nj} h_n$ be the transformed feature vector.

If the transformation matrix $T_{nj}$ exhibits symmetric or complementary weight patterns that cause significant weight cancellation, the difference between the transformed and original vectors, $||h'_n - h_n||_2$, will be insufficient to surpass a given anomaly detection threshold $\delta > 0$.
This proof is given in appendix A.

**Theorem 3:** Let $G = (V, E)$ represent an Ethereum transaction graph with $|V| = N$ nodes (accounts) and $|E| = M$ edges (transactions). Let $X \in \mathbb{R}^{N \times d}$ denote the feature matrix for the nodes, $A \in \mathbb{R}^{N \times N}$ the adjacency matrix representing transaction relationships, and $Y \in \{0, 1\}^N$ the binary labels indicating specific account behaviors. Probabilistic random walk sampling, defined by a sampling matrix $P$, improves the performance of a GCN for the task of predicting node labels $Y$ in the context of Ethereum networks.
This proof is given in appendix B.

# 3 Empirical Analysis

In this section, not only we provide details about the empirical analysis and evaluation methods, but also provide supporting information for readers to follow the experiments.

## 3.1 Datasets, Materials and Methods

We provide datasets and the code in the github link https://github.com/stefankam/temporal-spacial-anomaly-detection. We run the code on our department server running Linux equipped with a single GPU (NVIDIA A100 80GB PCIe), and 251Gi RAM.

Creating a complete transaction graph for all Ethereum blocks would be a computationally intensive task, as it would involve processing and storing a large amount of data. However, in the supplemental material we provide the code to generate a transaction history graph for a range of 100-1000 blocks. We further incorporate spatial and temporal node features to capture temporal aspects more explicitly:

**incoming_value_variance**: Variance of the transaction values received by the node. This metric quantifies the spread or dispersion of incoming transaction amounts, providing insight into the consistency or variability of funds received. **outgoing_value_variance**: Variance of the transaction values sent by the node. **activity_rate**: The activity rate of a node represents the total number of transactions (both incoming and outgoing) divided by the duration (in terms of blocks). It indicates the frequency of interactions involving the node over a specific period. **change_in_activity**: The change in activity refers to the difference in the number of transactions of the current block compared to the previous block for a given node. This metric captures fluctuations or deviations in transaction behavior over consecutive blocks. **time_since_last**: Time since the last transaction involving the node, measured as the difference between the current block number and the block number of the node's most recent transaction. It provides insights into the recency of activity associated with the node. **tx_volume**: Total transaction volume associated with the node, calculated as the sum of incoming and outgoing transaction values. This metric represents the overall magnitude of financial transactions involving the node. **frequent_large_transfers**: Indicator variable identifying addresses engaged

in frequent and large transfers. Nodes meeting specific thresholds for both transaction frequency and volume are flagged. **gas_price**: Additional feature relevant for MEV detection, representing the gas price paid for transactions. Gas price fluctuations can signal potential MEV activities such as frontrunning or transaction ordering strategies. **token_swaps**: Another feature for MEV detection, indicating involvement in token swaps or trades on decentralized exchanges (DEXs). Analysis of token swap transactions can reveal arbitrage opportunities or manipulative behavior by MEV bots. **smart_contract_interactions**: Feature identifying transactions interacting with known DeFi protocols or smart contracts. MEV bots may exploit vulnerabilities or manipulate protocol behaviors.

## 3.2  TRW-GCN combined method to detect anomalies

To apply graph convolutional layers to the blockchain data for aggregating information from neighboring nodes and edges, we'll use the PyTorch Geometric library. This library is specifically designed for graph-based data and includes various graph neural network layers, including graph convolutional layers. Note that training and testing a graph neural network on Ethereum dataset would require significant computational resources, as currently, the Ethereum network possesses about 20 million blocks, which are connected over the Ethereum network. In this study, we provide the transaction history within a specified range of 1000 blocks; we believe, adding blocks do not add any advantage.

In Algorithm 1, we intend to compare the anomaly detection of full- and sub-graphs (sampling using TRW). The graph convolution operation combines the features of neighboring nodes to update the representation of a given node. As node features, we input the 10 features indicated in 3.1 as vector representation; considering 20 hidden layers, 100 epochs, `lr=0.01`, `num_walks=10`, and `walk_length=100`, the resulting output vector aggregates information from all neighboring nodes. By using the nodes from TRW for training, the GCN will be more attuned to the time-dependent behaviors, leading to better detection of sudden spikes in transaction volume or unusual contract interactions that occur in quick succession. In our experiments, we employ TRW to sample nodes from the entire graph, ensuring that the graph's integrity is maintained. Here's how it can be done:

1. **Perform TRWs to Sample Nodes for Training:** The TRWs provide sequences of nodes representing paths through the Ethereum network graph. Nodes appearing frequently in these walks are often involved in recent temporal interactions.

2. **Train the GCN with the Sampled Nodes:** Instead of using the entire Ethereum network graph for training, use nodes sampled from the TRWs. This approach tailors the GCN to recognize patterns from the most temporally active parts of the Ethereum network.

Using the GCN with TRW combined method, one can achieve 1) Anomalies detected, 2) Training efficiency, and 3) Quality of embedding. The integration of TRW with GCNs offers a novel approach for generating embedding that capture both spatial and temporal patterns within the Ethereum network. These embedding are vital for understanding the underlying transaction dynamics and for effectively detecting anomalous activities. To evaluate the potential of the TRW-GCN methodology, we employ four distinct machine learning techniques: DBSCAN, SVM, Isolation Forest (IsoForest), and Local Outlier Factor (LOF). Wu et al. (12) indicated that they have obtained more than 500 million Ethereum addresses and 3.8 billion transaction records. However, only 1259 addresses are labeled as phishing addresses collected from EtherScamDB, which implies an extreme data imbalance as the biggest obstacle for phishing detection, therefore they used unsupervised learning detection method. We similarly use unsupervised learning for detection in our TRW-GCN algorithm.

The extensive use of these four diverse methods allows us to validate the efficacy of the TRW-GCN framework. The high anomaly detection rates in Figure 1 by clustering methods underscores the importance of algorithm selection. As easily observed, using the embedding generated by TRW-GCN in SVM method significantly improves anomaly detection, however other methods do not show any improvement in anomaly detection (averaged over 10 runs); the enhanced detection capabilities in SVM could be attributed to the TRW's ability to encapsulate temporal sequences and correlations of transactions. In Table 2, we compare these methods in terms of their precision, recall and F-score and compare with the outcome of SVM and IsoForest methods implemented in (12) (note that this paper focuses on Phishing detection in Ethereum Network, and is different from our dynamic approach in temporal anomaly detection). Our models are marking many data points as anomalies; precision stays relatively high, but low F-score. Algorithms like DBSCAN, LOF, Isolation Forest are unsupervised, so they often overpredict if not-cluster-fit noise is high or parameter tuning is off. DBSCAN is very sensitive to eps. Isolation Forest depends on contamination, and LOF is sensitive to n_neighbors.

| Algorithm 1: TRW- GCN combined method to detect anomalies |
|---|

**Steps:**
1. Load and Preprocess the graph $G$.
2. **For** each walk $k = 1$ to num_walks:
```
W = {w_k for k in range(1, num_walks+1) for
w_k in temporal_random_walk(k)}
// Aggregate walks in W
```
**End**
3. **For** training step:
```
F = torch.stack([f(v_i) for v_i in V], dim=0)
A = nx.to_numpy_matrix(graph, nodelist=V)
```
$M_{TRW}$, M = GCN(in_channels, hidden_channels, out_channels)
```
train(M_TRW, F, A) if use_TRW else train(M, F, A)
// Training using sampled-graphs
```
**End**
4. Apply DBSCAN, One-Class SVM, IsoForest, and LOF on embeddings from the trained GCN model $M$ to obtain anomalies.

| Algorithm 2: A Score-based anomaly detection associated with time-dependent behaviors |
|---|

**Steps:**
1. $G' = G(V, E)$ where $E$ has node attributes.
2. $X = [x_1, x_2, \ldots, x_n]$ for $n \in V$.
3. GCNModel with layers:
in_channels $\rightarrow$ hidden_channels $\rightarrow$ out_channels
4. TRW($G'$, start, length) returns walk $W$ and timestamps $T$
5. **For** each walk $i = 1$ to num_walks:
All_Walks $= \bigcup_{i=1}^{\text{num\_walks}}$ TRW($G'$, random_node, walk_length).
// Node Sampling via TRW
**End**
6. Node Frequency Computation:
$\text{freq}(v) = \frac{\text{occurrences of } v \text{ in All\_Walks}}{\text{max occurrences in All\_Walk}}$ for $v \in V$.
7. Anomaly Score Computation:
$S(v) = \frac{(\text{emb}(v)_{\text{latest}} - \mu(\text{emb}(v)))}{\sigma(\text{emb}(v))} \times \text{freq}(v)$
where emb is the node embedding, $\mu$ is the mean, and $\sigma$ is the standard deviation; anomalies are detected when S(v) > threshold $\delta$.

It is also interesting to find out which node features mainly contribute to anomaly detection; we show this in Figure 2. As illustrated by different colors, the feature 3-6 namely activity_rate, change_in_activity, time_since_last (mainly the temporal features) are the drivers of frequent anomalies (with dark blue colors), while tx_volume and frequent_large_transfers (with green colors) also produce anomalies but less frequently. Although we have obtained good insights into the method effectiveness to detect time-dependent patterns and features, but we should look for more precise and less prone to error detection method.

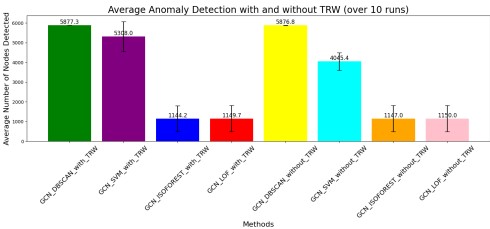

Figure 1: A comparison (mean, std) of 4 detection models namely dbscan, svm, isoforest and lof between full-graph and sub-graph with TRW sampling. Using TRW-GCN clearly improves SVM in anomaly detection; other methods do not seem to be improved.

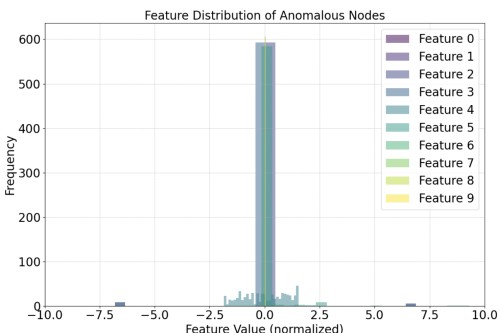

Figure 2: Feature distribution where Blue and Green colors: activity_rate, change_in_activity, and times_since_last show highest frequencies.

Table 2: Comparison of Precision/Recall/F-score of 4 methods with/out-TRW

| Method | Prec.(w-T) | Rec.(w-T) | F-S.(w-T) | Prec.(o-T) | Rec.(o-T) | F-S.(o-T) | Prec.(12) | Rec.(12) | F-S.(12) |
|---|---|---|---|---|---|---|---|---|---|
| DBSCAN | 0.799 | 0.485 | 0.604 | 0.799 | 0.485 | 0.604 | | | |
| SVM | 0.799 | 0.438 | 0.563 | 0.796 | 0.333 | 0.458 | 0.927 | 0.893 | 0.908 |
| IsoForest | 0.795 | 0.094 | 0.163 | 0.796 | 0.094 | 0.163 | 0.821 | 0.849 | 0.835 |
| LOF | 0.815 | 0.096 | 0.167 | 0.812 | 0.096 | 0.167 | | | |

## 3.3 Score-based anomaly pattern

While traditional methods compute anomaly scores based on the relative position or density of data points in the feature space, we need a method to be more focused on temporal dynamics, tracking the evolution of each node's embedding over time and weighing it by the node's frequency in the graph. To adapt the code to pick up anomalous patterns associated with time-dependent behaviors, the algorithm should be equipped to recognize such patterns. Hence, we augment the node features to capture recent activities with time features as explained in 3.1 dataset section, and after obtaining node embedding from the GCN, compute the anomaly score for each node based on its temporal behavior. The simplest way to achieve this is by computing the standard deviation of the node's feature over time and checking if the latest data point deviates significantly from its mean. This was initially discussed in Theorem 1, with weight cancellation argument in Theorem 2.

We explain all the steps in Algorithm 2. Initially, we define the node features to capture recent activities. After training the GCN and obtaining the embedding, we compute an anomaly score based on how much the recent transaction volume (the latest day in our case) deviates from the mean. We then use a visualization function to display nodes with an anomaly score beyond a certain threshold (in this case, we've used a z-score threshold of 2.0 which represents roughly 95% confidence).

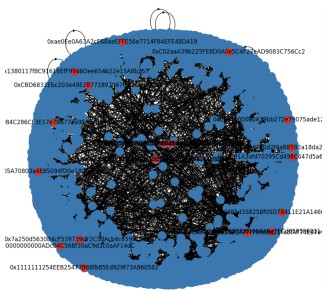 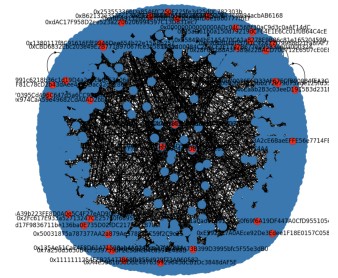

Figure 3: Anomaly detection in (left) 100 blocks with 6 features, (middle) 100 blocks with 10 features, (right) the anomalous addresses where the time-sensitive associated ones are hashed green.

In Figure 3, black points represent the vast majority of nodes in our specified Ethereum network dataset; they signify regular non-anomalous Ethereum addresses. Cluster of points inside and around the blue circle represent groupings of Ethereum addresses or contracts that have had frequent interactions with each other. The density or proximity of points to each other indicates how closely those addresses or contracts are related. Red points would represent the nodes that have been flagged as anomalous based on their recent behavior. The code identifies them by computing an anomaly score, and those exceeding a threshold are colored red. In the left graph, there are just 20 nodes detected as anomaly in 100 blocks where we used 6 structural features in our detection algorithm, while in the middle graph, we used 10 features to detect anomalies in the same 100 blocks, and 12 more suspicious addresses are detected, hashed in green in the right figure. This signifies the importance of temporal feature selection, as by adding 4 temporal features we would be able to detect missing anomalies. We checked these addresses in Ether explorer website https://etherscan.io , and found the corresponding labels such as MEV Bot, Metamask: Swap, Uniswap, Wrapped Ether, Rollbit, Blur: Bidding, which are mainly time-sensitive transactions or contracts, see next section for explanation on what is normal versus anomaly. In Table 3, we explain the types of such detected anomalies and the associated addresses. This is a proof of cross-checking with the ground-truth .

Table 3: Some types of detected anomalies

| Ethereum addresses for anomalies detected from Figure 3 | Ground Truth : cross-check with https://etherscan.io/ |
| --- | --- |
| 0x6F1cDbBb4d53d226CF4B917 bF768B94acbAB6168 | MEV Bot; certain activities may be considered harmful |
| 0x3fC91A3afd70395Cd496C647 d5a6CC9D4B2b7FAD | Uniswap (users to swap various ERC-20 tokens) |
| 0x881D40237659C251811CEC9 c364ef91dC08D300C | Metamask Swap router |
| 0x0000000000A39bb272e79075a de125fd351887Ac | Flashloan; Detecting involves transactions with large token volumes |

Table 4: TRW-GCN versus TGAT for eth_latest_100_block file, and z-score threshold of 2.0.

| Model | Accuracy / # Anomalies detected |
| --- | --- |
| TRW-GCN | 94.5% / 20 |
| TGAT | 85.3% / 23 |

## 3.4 Normal versus Anomaly, Baseline algorithm, Algorithm complexity, and the Ground truth

In Ethereum, what may be considered normal or anomalous behavior can vary depending on various factors such as market conditions, network activity, and the specific use cases of different addresses or smart contracts. Time-sensitive irregularities in Ethereum transactions refer to anomalies that occur within specific time frames or exhibit patterns that are indicative of immediate or rapid actions. These irregularities may include instances of rapid buying or selling of assets, front-running other traders, MEV activities, flash loan exploits, or token swaps executed within short time intervals. Identifying these irregularities requires analyzing transactional data in real-time or within narrow time windows to capture anomalous behaviors as they occur. See Table 5 for a list of time-sensitive items in Ethereum network including transactions, contracts, and platform activities. Our objective is to identify such instances; upon identifying suspicious transactions, our approach advocates for

further investigation. In Table 3, we cross-reference the transaction details with etherscan.io (which represents a source for ground truth, where one finds more information about an anomaly).

Table 5: some time sensitive items on Ethereum network and their definitions

| Time sensitive items | Definitions |
| --- | --- |
| MEV Bot | MEV refers to the additional profit that miners can extract from the Ethereum network by reordering, censoring, or including transactions in blocks. The timing of transactions and block mining can affect the potential profit extracted by MEV bots. MEV can affect fairness and efficiency of the Ethereum network. |
| Metamask: Swap Uniswap | Uniswap is a decentralized exchange (DEX) protocol on Ethereum, and swaps conducted through MetaMask can be time-sensitive, especially considering the volatility of cryptocurrency prices and liquidity on Uniswap. |
| Flashloan | Flash loans are uncollateralized loans that must be borrowed and repaid within a single transaction block. These loans are often used for arbitrage, liquidations, or other trading strategies that require rapid execution. |
| Wrapped Ether (WETH) | Wrapped Ether (WETH) is an Ethereum token pegged to the value of Ether (ETH). Transactions involving WETH can be time-sensitive, especially if they're related to trading, liquidity provision, or token swaps. |
| Token Launches and Airdrops | Token launches and airdrops often have predefined distribution schedules or timeframes during which users can claim or receive tokens. |
| Smart Contract Exploits | Exploiting vulnerabilities in smart contracts often requires precise timing to execute malicious transactions before vulnerabilities are patched or mitigated. |

Similar to the papers by Wu et al. (12), Zhang et al. (16), and Feng et al. (17), as baseline algorithms for comparison, common unsupervised methods such as Isolation Forest, One-Class SVM, LOF and DBSCAN are employed. Evaluation metrics, including precision, recall, F1 score in Table 2 are utilized to assess the performance of the proposed methods. However, clustering methods report many anomalies; DBSCAN, If eps is too small, leads to many points treated as noise. LOF also depends heavily on n_neighbors, and Isolation Forest depends on contamination parameter. That is why the study further introduces a statistically-based scoring method to identify anomalous nodes. The scoring function employs different z-score thresholds of 1.0, 1.5, and 2.0 (95% confidence level). Furthermore, we compare the results obtained from our scoring method with the ground truth on etherscan.io, providing a case-by-case evaluation of detected time-sensitive anomalies in Table 3.

We further compare the TRW-GCN model against the state-of-the-art TGAT method. TGAT is specifically designed to incorporate temporal information through time-aware positional encodings and attention mechanisms. However, in practice, our experiments revealed significant computational and performance challenges when applying TGAT, particularly in complex, high-frequency networks such as Ethereum. TGAT's multi-head attention mechanism introduces substantial overhead due to repeated matrix multiplications and attention score computations. Additionally, its dependency on fine-grained temporal edge attributes adds complexity to both preprocessing and model execution, resulting in long training time and memory inefficiency. In contrast, TRW-GCN's use of temporal random walks allows it to construct meaningful local temporal subgraphs with controlled depth and temporal relevance, making it significantly more scalable without sacrificing temporal fidelity. From a performance standpoint, TGAT achieved an accuracy of 85.3% detecting 23 anomalies, while our TRW-GCN model — coupled with a scoring classifier — has achieved an average accuracy of 94.5% detecting 20 anomalies, see Table 4. One likely factor behind this discrepancy is TGAT's sensitivity to the temporal quality and distribution of data. In Ethereum, where transactions are bursty and user behavior is non-uniform, TGAT struggles to generalize effectively. Moreover, TGAT's reliance on explicit node identities (e.g., blockchain addresses) complicates indexing and neighborhood retrieval, especially in networks with millions of ephemeral or sparsely active nodes. TRW-GCN, in contrast, is more robust in such settings due to its walk-based sampling, which implicitly encodes temporal structure without depending on densely connected or temporally smooth interactions.

## 4   Conclusion

The evolution and complexity of the Ethereum network has heightened the urgency for temporal anomaly detection methods. Through our research, we've demonstrated that the combined TRW-GCN methed offers a solution to this challenge. This fusion has enabled us to delve deeper into the intricate spatial-temporal patterns of Ethereum transactions, offering a refined lens for anomaly detection. We have shown the methodology usefulness by expressing and proving three distinct theorems, full empirical analysis and evaluation. While this approach is used to obtain the embedding, we have compared different clustering and scoring classification methods to obtain highest precision in anomaly detection, and verified with the ground truth found on etherscan.io. Furthermore, we have demonstrated that the TRW-GCN method improves anomaly detection versus the state-of-the-art TGAT method, also proved how probabilistic sampling improves GCN performance in Appendix B.

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

# A   Significant weight cancellation

**Theorem 2:** Let $\mathbb{R}^m$ be a vector space, and let $h_n \in \mathbb{R}^m$ represent a feature vector. Define a temporal transformation matrix $T_{nj} \in \mathbb{R}^{m \times m}$, where each entry $t_{ij}$ encodes the temporal weights. Let $h'_n = T_{nj} h_n$ be the transformed feature vector.

If the transformation matrix $T_{nj}$ exhibits symmetric or complementary weight patterns that cause significant weight cancellation, the difference between the transformed and original vectors, $||h'_n - h_n||_2$, will be insufficient to surpass a given anomaly detection threshold $\delta > 0$. Specifically, weight cancellation occurs if:

$$\sum_{j=1}^{m} t_{ij} h_{nj} \approx h_{ni}, \quad \forall i \in \{1, 2, \ldots, m\}. \tag{10}$$

**Proof:**

The transformed feature vector $h'_n = T_{nj} h_n$ can be expressed component-wise as:

$$h'_{ni} = \sum_{j=1}^{m} t_{ij} h_{nj}, \quad \forall i \in \{1, 2, \ldots, m\}. \tag{11}$$

The Euclidean norm of the difference between the transformed and original feature vectors is given by:

$$||h'_n - h_n||_2 = \sqrt{\sum_{i=1}^{m} \left( \sum_{j=1}^{m} t_{ij} h_{nj} - h_{ni} \right)^2}. \tag{12}$$

For $||h'_n - h_n||_2 > \delta$, the inequality must hold:

$$\sum_{i=1}^{m} \left( \sum_{j=1}^{m} t_{ij} h_{nj} - h_{ni} \right)^2 > \delta^2. \tag{13}$$

This implies that, for at least one $i$, the inner term $\sum_{j=1}^{m} t_{ij} h_{nj} - h_{ni}$ must be at least $\delta^2$. Therefore, $T_{nj}$ must introduce a significant alteration to the distribution of $h_n$. Weight cancellation occurs when $T_{nj}$ has structural properties that lead to minimal change in $h_n$. Consider the following cases:

- **Symmetry in** $T_{nj}$: If $T_{nj}$ is symmetric ($t_{ij} = t_{ji}$) and $h_n$ has symmetric properties, the transformation may yield:

$$\sum_{j=1}^{m} t_{ij} h_{nj} \approx h_{ni}, \quad \forall i. \tag{14}$$

In this scenario, the transformed feature vector $h'_n$ closely resembles original vector $h_n$, leading to

$$||h'_n - h_n||_2 \approx 0. \tag{15}$$

- **Complementary Weights**: If $T_{nj}$ contains complementary weights, such that certain entries $t_{ij}$ and $t_{ik}$ satisfy $t_{ij} + t_{ik} = 0$, and if $h_{nj} \approx h_{nk}$, then the contributions from $h_{nj}$ and $h_{nk}$ cancel each other out:

$$\sum_{j=1}^{m} t_{ij} h_{nj} \approx 0, \quad \text{for certain } i. \tag{16}$$

- **Spectral Properties of** $T_{nj}$: If $T_{nj}$ has eigenvalues close to 1, it behaves similarly to an identity matrix, resulting in $h'_n \approx h_n$. Orthogonality in rows or columns of $T_{nj}$ may also preserve the magnitude of $h_n$, leading to minimal changes in $h'_n$.

In scenarios where weight cancellation occurs, the transformation $T_{nj}$ fails to introduce meaningful changes to the feature vector $h_n$. Consequently, anomalies influenced by temporal factors may not be detectable, as the difference $||h'_n - h_n||_2$ remains below the threshold $\delta$.

## B  Improvement of GCN performance with probabilistic sampling

**Theorem 3:** Improvement of GCN performance with probabilistic sampling in the context of random walk sampling.

Consider a simplified Ethereum transaction graph with N accounts (nodes), and M transactions (edges) between them. Prove the performance improvement of a GCN in terms of loss, using probabilistic sampling for the task of predicting account behaviors, considering the following assumptions:
1. Nodes (accounts) have features represented by vectors in a feature matrix X.
2. The adjacency matrix A represents transaction relationships between accounts.
3. Binary labels Y indicate specific account behaviors.

**Proof.**

## B.1 Traditional GCN performance

Start with the definition of the normalized graph Laplacian $L = I - D^{-\frac{1}{2}} A D^{-\frac{1}{2}}$, where $D$ is the diagonal degree matrix and $A$ is the adjacency matrix.

Derive the eigenvalues and eigenvectors of the Laplacian matrix L and show their significance in capturing graph structure. Derive the performance of a GCN trained on the full graph using these eigenvalues and eigenvectors:

Step 1: Deriving Eigenvalues and Eigenvectors of the Laplacian matrix $L$

Given the normalized graph Laplacian matrix $L$, let $\lambda$ be an eigenvalue of $L$ and $v$ be the corresponding eigenvector. In the equation $Lv = \lambda v$, solving for $\lambda$ and $v$, we get:

$$D^{-\frac{1}{2}} A D^{-\frac{1}{2}} v = (1 - \lambda) v \tag{17}$$

$$A D^{-\frac{1}{2}} v = (1 - \lambda) D^{\frac{1}{2}} v \tag{18}$$

This equation implies that $D^{-\frac{1}{2}} A D^{-\frac{1}{2}}$ is a symmetric matrix that is diagonalized by the eigenvectors $v$ with corresponding eigenvalues $1 - \lambda$. The eigenvectors and eigenvalues of $L$ capture the graph's structural information. Larger eigenvalues correspond to well-connected clusters of nodes in the graph, while smaller eigenvalues correspond to isolated groups or individual nodes.

Step 2: Deriving GCN performance using eigenvalues and eigenvectors

Now let's consider a scenario where we're using a GCN to predict node labels (such as predicting high-value transactions) on the full graph. The GCN's propagation rule can be written as:

$$h^{(l+1)} = f(\hat{A} h^{(l)} W^{(l)}) \tag{19}$$

where $h^{(l)}$ is the node embedding matrix at layer $l$, $f$ is an activation function, and $\hat{A} = D^{-\frac{1}{2}} A D^{-\frac{1}{2}}$. is the symmetrically normalized adjacency matrix, and $W^{(l)}$ is the weight matrix at layer l. The key insight is that if we stack multiple GCN layers, the propagation rule becomes:

$$\begin{aligned} h^{(L)} &= f(\hat{A} h^{(L-1)} W^{(L-1)}) \\ &= f(\hat{A} f(\hat{A} h^{(L-2)} W^{(L-2)}) W^{(L-1)}) \dots \end{aligned} \tag{20}$$

We can simplify this as:

$$h^{(L)} = f\left( \hat{A}^{(l)} h^{(0)} W^{(0)} \prod_{l=1}^{L-1} W^{(l)} \right) \tag{21}$$

Using the spectral graph theory, we know that $\hat{A}^{(l)}$ captures information about the graph's structure up to L-length paths. The eigenvalues and eigenvectors of $\hat{A}^{(l)}$ indicate the influence of different sampled-graphs of length L on the node embeddings.

## B.2 Probabilistic Sampling Approach

In this step, we'll introduce a probabilistic sampling strategy to select a subset of nodes and their associated transactions. This strategy aims to prioritize nodes with certain characteristics or properties, such as high transaction activity or potential involvement in high-value transactions. Assign a probability $p_i$ to each node i based on certain characteristics. For example, nodes with higher transaction activity, larger balances, or more connections might be assigned higher probabilities. For each node i, perform a random sampling with probability $p_i$ to determine whether the node is included in the sampled subset. Consider a graph with $N$ nodes represented as $N = \{1, 2, \dots, N\}$. Each node $i$ has associated characteristics described by a feature vector $\mathbf{X}_i = [X_{i,1}, X_{i,2}, \dots, X_{i,k}]$, where $K$ is the number of characteristics. Define the probability $p_i$ for node $i$ as a function of its feature vector $\mathbf{X}_i$: $p_i = f(\mathbf{X}_i)$. Here, $f(\cdot)$ is a function that captures how the characteristics of node $i$ are transformed into a probability. The specific form of $f(\cdot)$ depends on the characteristics and the desired probabilistic behavior. For example, $f(\mathbf{X}_i)$ could be defined as a linear combination of the elements in $\mathbf{X}_i$:

$$p_i = \sum_{j=1}^{K} \omega_j X_{i,j} \tag{22}$$

Where $\omega_j$ are weights associated with each characteristic. The weights $\omega_j$ can be used to emphasize or downplay the importance of specific characteristics in determining the probability. After obtaining $p_i$ values for all nodes, normalize them to ensure they sum up to 1. Nodes with higher normalized probabilities are more likely to be included in the sampled subset.

$$p_{\text{normalized}} = \frac{p_i}{\sum_{j=1}^{N} p_j} \tag{23}$$

## B.3 Graph Laplacian for Sampled Graph

Given the sampled adjacency matrix $\hat{A}_{\text{sampled}}$, we want to derive the graph Laplacian $\hat{L}_{\text{sampled}}$ for the sampled graph. The graph Laplacian $\hat{L}_{\text{sampled}}$ is given by:

$$\hat{L}_{\text{sampled}} = I - \hat{D}_{\text{sampled}}^{-\frac{1}{2}} \hat{A}_{\text{sampled}} \hat{D}_{\text{sampled}}^{-\frac{1}{2}} \tag{24}$$

Where $\hat{D}_{\text{sampled}}$ is the diagonal degree matrix of the sampled graph, where each entry $d_{ii}$ corresponds to the degree of node i in the sampled graph, and $\hat{A}_{\text{sampled}}$ is the sampled adjacency matrix.

$$d_{ii} = \sum_{j=1}^{N_{\text{sampled}}} \hat{A}_{\text{sampled},ij} \tag{25}$$

The modified Laplacian captures the structural properties of the sampled graph and is essential for understanding its graph-based properties. As eigenvalues of the sampled graph, we derive

$$\hat{L}_{\text{sampled}} = I - \hat{D}_{\text{sampled}}^{-\frac{1}{2}} \hat{A}_{\text{sampled}} \hat{D}_{\text{sampled}}^{-\frac{1}{2}} \tag{26}$$

as the normalized graph Laplacian for the sampled graph. Let $\hat{\lambda}_i$ be the $i$-th eigenvalue of $\hat{L}_{\text{sampled}}$ and $\hat{v}_i$ be the corresponding eigenvector. We have

$$\hat{L}_{\text{sampled}} \hat{v}_i = \hat{\lambda}_i \hat{v}_i \tag{27}$$

The goal is to compare the eigenvalues of $L$ with the eigenvalues of $\hat{L}_{\text{sampled}}$ and show convergence under certain conditions. As the sample size $N_{\text{sampled}}$ approaches the total number of nodes $N$ in the original graph, $\hat{L}_{\text{sampled}}$ converges to $L$. Eigenvalues of $\hat{L}_{\text{sampled}}$ converge to the eigenvalues of $L$.

## B.4 Impact on GCN Performance

To demonstrate that the performance $E_{\text{sampled}}$ of a GCN on a sampled graph, is greater than or equal to the performance $E_{\text{full}}$ on the full graph, we use two approaches:

**1. Reduction of Noise and Retention of Structural Information**

The total loss $\mathcal{L}$ of a GCN can be expressed as:

$$\mathcal{L}(h) = \mathcal{L}_{\text{train}}(h) + \mathcal{E}(h) \tag{28}$$

where:

- $\mathcal{L}_{\text{train}}(h)$: Loss on the training set.
- $\mathcal{E}(h)$: Generalization error (e.g., noise or overfitting effects).

For the sampled graph $G_{\text{sampled}}$, the loss becomes:

$$\mathcal{L}(h_{\text{sampled}}) = \mathcal{L}_{\text{train}}(h_{\text{sampled}}) + \mathcal{E}(h_{\text{sampled}}) \tag{29}$$

Probabilistic sampling prioritizes nodes with higher relevance (e.g., higher degree or centrality) by assigning sampling probabilities $p_i$:

$$p_i = f(X_i), \quad p_{\text{normalized}} = \frac{p_i}{\sum_j p_j} \tag{30}$$

where $X_i$ represents node features. By emphasizing relevant nodes, noise is reduced, and:

$$\mathcal{E}(h_{\text{sampled}}) < \mathcal{E}(h) \tag{31}$$

Thus, the total loss on the sampled graph satisfies:

$$\mathcal{L}(h_{\text{sampled}}) < \mathcal{L}(h) \tag{32}$$

**2. Reduction in Computational Complexity and Faster Convergence**

The computational complexity of a GCN is:

$$\mathcal{O}(L \cdot (N + M) \cdot d^2) \tag{33}$$

where $N$ is the number of nodes, $M$ is the number of edges, $L$ is the number of layers, and $d$ is the embedding dimension. For the sampled graph $G_{\text{sampled}}$, the complexity reduces to:

$$\mathcal{O}(L \cdot (N_{\text{sampled}} + M_{\text{sampled}}) \cdot d^2) \tag{34}$$

Since $N_{\text{sampled}} \ll N$ and $M_{\text{sampled}} \ll M$, the sampled graph enables faster convergence. Let the convergence rate $R$ be inversely proportional to the size of the graph:

$$R(G_{\text{sampled}}) > R(G) \tag{35}$$

Thus, the sampled graph converges faster and reaches a better minimum of the loss function:

$$\mathcal{L}(h_{\text{sampled}}) \text{ decreases faster compared to } \mathcal{L}(h) \tag{36}$$

Given the reduced noise, retention of structural information, and faster convergence, probabilistic sampling ensures that:

$$E_{\text{sampled}} > E_{\text{full}} \tag{37}$$

### B.5 How TRW impacts on GCN performance as compared to traditional sampling

Let's delve into empirical justification on why TRW sampling could enhance the performance of GCNs, especially in temporal networks like Ethereum. For a detailed mathematical proof on the probabilistic sampling in GCN, you are invited to read appendix B1-B4. One issue with traditional random walks is the potential for creating "jumps" between temporally distant nodes, breaking the temporal consistency. GCNs rely on the local aggregation of information, and since TRW promotes smoother temporal signals, GCNs can potentially learn better node representations. Temporal consistency ensures that the sequences are logically and temporally ordered. This can be crucial for predicting future events or understanding time-evolving patterns, making GCNs more reliable. We compare different GCN models (including graphSAGE and graph attention network GAT model) for fullgraph, and sampled-graph with traditional and temporal random walk in Figure 4. Although one sees little difference between the accuracy of the fullgraph and the sampled-graph in graphSAGE and GAT models (18), one can see that traditional random walk and temporal random walk improve GCN accuracy, where TRW shows even further improvement than the traditional random walk.

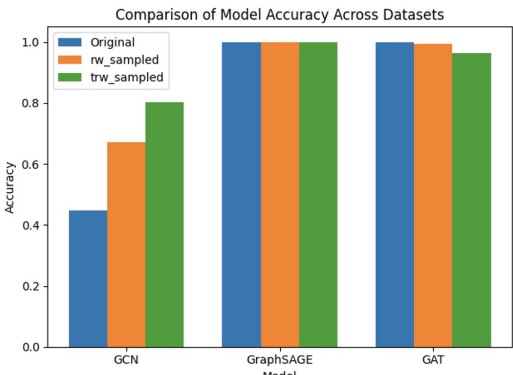

Figure 4: Comparison of fullgraph, traditional RW and TRW-based on sampled graph in 100 blocks.

# C Checklist Responses

1. **Claims:** *Yes.* The abstract and introduction reflect the contributions of the paper. TRW-GCN is proposed as a domain-specific temporal GCN variant tailored to Ethereum transaction networks. The use of probabilistic temporal walks and their effect on anomaly detection are experimentally demonstrated. The paper acknowledges that the model is not intended as a general-purpose method, and the scope is clearly limited to complex blockchain structures.

2. **Limitations:** *Yes.* Limitations are discussed in the text. Notably, the model is tailored to Ethereum-like graphs and may not generalize to all temporal graph domains. Limitations in comparison scope (e.g., AddGraph, TGAT) and reliance on temporal features that may be noisy are acknowledged. We also observed that TGAT results in higher computational costs, primarily due to its multi-head attention mechanism, which involves multiple passes of matrix multiplications and attention score computations. Furthermore, TGAT's reliance on temporal edge attributes added another layer of complexity, further increasing the computational burden.

3. **Theory, Assumptions and Proofs:** *Yes.* We provided all theoretical claims, stated all assumptions clearly before theorem statements, and provided formal proofs either in the main paper or appendix.

4. **Experimental Result Reproducibility:** *Yes.* Both data and code are attached at submission which also explains how to obtain the paper results.

5. **Open Access to Data and Code:** *Yes.* Full Ethereum dataset is publicly available; nevertheless, we provide our created dataset and the code in the github link https://github.com/stefankam/temporal-spacial-anomaly-detection, which is anonymized.

6. **Experimental Setting/Details:** *Yes.* Full training and testing splits, model hyperparameters, walk lengths, and walk counts are provided in the text. Comparisons with TGAT and other unsupervised methods (e.g., SVM, ISOForest) are described.

7. **Experiment Statistical Significance:** *Yes.* The scoring method is based on z-score thresholds (1.0, 1.5, 2.0), corresponding to standard confidence levels (e.g., 95%). The reported precision/recall/f1 are averaged over multiple thresholds and visualized. Confidence intervals are also included in Figure 1.

8. **Experiments Compute Resource:** *Yes.* Experiments were run on our department server running Linux equipped with a single GPU (NVIDIA A100 80GB PCIe), and 251Gi RAM..

9. **Code of Ethics:** *Yes.* The research conforms to NeurIPS Code of Ethics. No human or sensitive data was used. All datasets are public and open.

10. **Broader Impacts:** *Yes.* The paper discusses anomaly detection and classification systems. Limitations of false positives are acknowledged specially in the clustering methods like dbscan which demonstrate high number of anomalies' detection. Future work could help mitigate misclassification risks, and further automation.

11. **Safeguards:** *N/A.* No pretrained models with dual-use risks are released. The framework is domain-specific and does not apply to general-purpose generative tasks.

12. **Licenses:** *Yes.* Ethereum transaction data is public and under open access. All reused datasets (e.g., etherscan.io) are cited appropriately. Libraries used include PyTorch Geometric (MIT License).

13. **Assets:** *No.* While no new datasets are introduced, the model artifacts and scripts will be documented and released.

14. **Crowdsourcing and Research with Human Subjects:** *N/A.* No human data or crowdsourcing was involved.

15. **IRB Approvals:** *N/A.* Not applicable as no human or user-generated content was analyzed.

16. **Declaration of LLM Usage:** *Yes.* LLMs (e.g., ChatGPT) were used only for editing and understanding of some technical concepts. They did not influence model design or methodology.

