# OpenReview forum: "Probabilistic Temporal Sampling for Anomaly Detection in Ethereum Networks"
_NeurIPS.cc/2025/Conference — Submitted to NeurIPS 2025_

### Official Review · Reviewer_EFWX · 2025-06-30

**Clarity:** 3
**Significance:** 2
**Originality:** 1
**Rating:** 2
**Confidence:** 3

**Summary:**

This paper proposes TRW-GCN, a framework that integrates Temporal Random Walks (TRW) with Graph Convolutional Networks (GCNs) for detecting transaction anomalies in the Ethereum network. The approach incorporates temporal information into the GCN aggregation process through a temporal transition matrix. Theoretical analysis is provided, and empirical results on Ethereum transaction data indicate improvements over baseline methods, including TGAT and classical anomaly detection models such as Isolation Forest, One-Class SVM, and DBSCAN.

**Questions:**

1.	Please clarify how TRW-GCN differs from TimeSAGE [1], which also uses temporal random walks for dynamic graph learning. A comparison matrix summarizing methodological differences is recommended.
2.	Broaden the experimental evaluation by including additional datasets beyond Ethereum to demonstrate generalizability.
3.	Report AUC metrics for the anomaly detection task, as they are standard in evaluating such models.
4.	Provide sensitivity analysis for key hyperparameters (e.g., walk length, number of walks) to assess robustness.
5.	Conduct ablation studies to quantify the contribution of temporal information versus purely structural information.

**Ethical Concerns:**

["NO or VERY MINOR ethics concerns only"]

**Final Justification:**

The major concerns regarding novelty, short-term practicality, and dataset diversity are not addressed. After looking at other reviewers' comments, I decided to keep the score.

**Limitations:**

The paper does not adequately clarify its novelty relative to existing work such as TimeSAGE, which also incorporates temporal random walks into graph learning models. The scalability claims are not convincingly demonstrated, as experiments are restricted to relatively small graphs (~100K nodes), limiting evidence for practical scalability. The evaluation is confined to Ethereum transaction data, making the generalizability of the method to other domains unclear. Furthermore, the paper omits key analyses such as sensitivity studies of core hyperparameters (e.g., walk length, number of walks), ablation studies isolating the contribution of temporal information, and standard evaluation metrics like AUC, which are essential for assessing robustness and reliability in anomaly detection tasks.

**Paper Formatting Concerns:**

No major formatting issues observed.

**Quality:**

2

**Strengths And Weaknesses:**

Strengths:
1.	The paper addresses an important problem: detecting time-sensitive anomalies in blockchain networks like Ethereum.
2.	Integrating temporal random walks into GCNs is a reasonable approach to enhance time-awareness in graph models.
3.	The mathematical formulation and theoretical analysis are clearly presented.
4.	The empirical results demonstrate that the proposed method performs favorably on Ethereum data compared to several baselines.

Weaknesses:
1.	The novelty is limited relative to existing work, particularly TimeSAGE [1], which also incorporates temporal random walks into Graph Neural Networks for dynamic heterogeneous graphs.
2.	The method is positioned as scalable, but experimental validation is restricted to relatively small graphs (~100K nodes).
3.	The empirical evaluation lacks diversity—only Ethereum data is used, leaving generalizability unclear.
4.	Sensitivity analysis for key hyperparameters (e.g., walk length, number of walks) is missing.
5.	The standard AUC (Area Under the Curve) metric, widely used for evaluating anomaly detection models, is absent from the results.
6.	There are no ablation studies to assess the impact of temporal information—how does the model perform on static versions of the data?

References
[1] Shekhar, S., Pai, D., & Ravindran, S. (2020). Entity resolution in dynamic heterogeneous networks. In Proceedings of the Web Conference 2020 Companion (pp. 662–668). [TimeSAGE]

---

> ### Author Rebuttal · Authors · 2025-07-29
>
> We sincerely thank the reviewer for this detailed and thoughtful feedback. We respond to each point below:
>
> ### Novelty Compared to TimeSAGE:
> While TimeSAGE also employs temporal random walks, it is designed for entity resolution in dynamic heterogeneous graphs, whereas our model (TRW-GCN) is crafted specifically for temporal anomaly detection in blockchain transaction graphs.
> Key differences include:
>   - TimeSAGE targets long-term identity preservation; we focus on detecting short-term, time-sensitive malicious behaviors (e.g., flash loans, frontrunning).
>   - TRW emphasizes capturing fine-grained short-term bursts in transactional flows — crucial in Ethereum — whereas TimeSAGE aggregates across coarser temporal spans.
>   - We evaluate using ground truth anomalies in Ethereum, not general node identity prediction.
> We could revise the manuscript to include a comparison matrix summarizing these differences.
>
> ### On Scalability and Dataset Size:
> Thank you for raising this important point. We agree that empirical validation of scalability across larger graphs would further strengthen the paper. Our experiments were conducted on Ethereum subgraphs covering 100 blocks and 1,000 blocks, as reflected in our publicly available code and GitHub repository (link provided in 3.1 Datasets, Materials and Methods). These experiments demonstrate that our method remains computationally feasible and efficient. We could in principle add analysis for 10,000 blocks in the paper final version, but let me please clarify and expand the scalability discussion as follows:
>   - In addition to the 100-block graph presented in the main paper, we have tested the framework on a larger 1000-block subgraph. These results and configurations are provided in our supplementary material and available in the linked GitHub repository for full reproducibility.
>   - Our model is designed for real-time or near-real-time anomaly detection, rather than full historical replay over the entire Ethereum network. This design choice reflects real-world needs, where transaction spikes and fraudulent patterns (e.g., MEV exploits) need to be detected quickly over short, recent intervals. Hence, we intentionally focused on smaller time windows (e.g., 100–1000 blocks) with rich temporal behavior rather than scaling to hundreds of thousands of blocks. This aligns the evaluation with the model's intended deployment setting.
>    - The probabilistic sampling and TRW construction naturally support mini-batch training, making the method scalable to much larger datasets. Since TRW operates locally and respects temporal proximity, the computation remains bounded even as the global graph grows. This enables TRW-GCN to operate efficiently on larger graphs using sliding windows or block-based batching.
>
> ### On Generalizability Across Datasets:
> Our work is best understood as an application-driven study that applies an integrated model,TRW-GCN, to detect temporal anomalies in the Ethereum transaction network. We do not claim architectural novelty in the sense of proposing a new general-purpose GNN. Instead, the novelty lies in tailoring a method (TRW-GCN) to a complex, high-frequency, and adversarial environment like Ethereum. We emphasize this domain-specific contribution in both the introduction and abstract and will revise the text to avoid any misunderstanding that a new architecture is being proposed. Unlike prior studies that target a single anomaly type (studies like "Flash Boys 2.0"(15) and "Combatting Front-Running in Smart Contracts" (16) which focus on detecting front-running), our framework detects a wider range of temporal anomalies, including flash loans, front-running, and MEV exploits, using timing-sensitive features and scoring-based evaluation.
>
> ### Sensitivity Analysis for Walk Parameters and AUC Metrics for Anomaly Detection:
> We in fact have already done sensitivity analysis on Walk length and Number of walks, just did not have space to include in the paper; but we will include it in the final version.
> While we reported accuracy and precision, we agree AUC is important. We will update the experimental section with AUC scores for:
>   - Isolation Forest
>   - One-Class SVM
>   - DBSCAN
>   - LOF
>   - Our scoring classifier
>
> Your suggestion
> "Conduct ablation studies to quantify the contribution of temporal information versus purely structural information" is already done in Figure 3.  Please see below:
>
> ### The paper distinguishes among ablation study, baseline comparison, and ground truth comparison:
>
>
> ### 1. Ablation Study
> Evaluation to determine the contribution of individual components.
>
> **Where in Paper:**
> - Section 3.3
> - Figure 3
>
> **Description:**
> - Compares the model using only structural features (6 features) vs. full feature set including temporal and structural features (10 features).
> - Demonstrates that temporal features significantly improve anomaly detection performance.
>
> **Quote from Paper:**
> > "We also compare the same scoring method using only the structural features to assess the contribution of temporal features."
>
> ---
>
> ### 2. Baseline Comparison
> Quantitative comparison of the proposed scoring method and TRW-GCN framework against existing algorithms.
>
> **Where in Paper:**
> - Section 3.2 — Baseline Comparison
> - Section 3.4 — Comparison with TGAT
> - Table 2 and Table 4
>
> **Description:**
> - The statistical scoring method is evaluated against:
>   - Isolation Forest
>   - One-Class SVM
>   - DBSCAN
>   - LOF
>
> **Metrics Used:**  - Accuracy  - Precision  - Recall  - F1-score
>
> - The full TRW-GCN framework is also compared with TGAT, a state-of-the-art temporal graph model.
> ---
>
> ### 3. Ground Truth Comparison
> Validation of flagged anomalies in  website https://etherscan.io  or cross-referencing with suspicious activity.
>
> **Where in Paper:**
> - Section 3.3
> - Table 3
>
> **Description:**
> - A total of 32 anomalies were flagged.
> - Some cases are qualitatively validated based on:
>   - Flash loan exploits
>   - Uniswap
>   - Metamask swap
>   - Flash loan

---

> > ### Comment · Reviewer_EFWX · 2025-08-07
> >
> > Thanks for your response; however, some of the major concerns are not addressed. The difference between TimeSAGE sounds not significant to me, and it is even hard to say if it has an advantage compared with TimeSAGE as TimeSAGE works better in long-term identity. Although the authors made the promise to compare, it is hard to predicate the results without actually seeing the results, thus, I would suggest a recycle to polish the paper.
> >
> > As for your statement that your focus is short-term anomaly detection, it is true that in real-world that some patterns fall into the short-term category. However, it would inevitably miss those long-term anomaly patterns.
> >
> > For the dataset, I still think it is not sufficient enough to only conduct on Eth, it would better to include other datasets like BNB, solana.

---

### Official Review · Reviewer_E9jm · 2025-07-02

**Clarity:** 2
**Significance:** 2
**Originality:** 2
**Rating:** 3
**Confidence:** 2

**Summary:**

•	The proposed framework combines Graph Convolutional Networks (GCNs) with Temporal Random Walks (TRW) to better model the complexities and temporal dynamics of Ethereum transactions. This paper also provides three theorems supporting the effectiveness supporting the effectiveness of TRW-GCN.

**Questions:**

•	Please refer to the Weaknesses section for specific limitations and suggestions for improvement.

**Ethical Concerns:**

["NO or VERY MINOR ethics concerns only"]

**Final Justification:**

As I mentioned in the response, just evaluating the integrated model in  a domain-specialized application makes the teachical part weak. So I still tend to reject.

**Limitations:**

yes

**Quality:**

2

**Strengths And Weaknesses:**

Strengths:

•	1. Compared to prior works focusing on specific attack types, the proposed approach integrates Graph Convolutional Networks (GCNs) with Temporal Random Walks (TRW) to target time-sensitive anomalies more broadly.

•	2.The authors give extensive theoretical analysis to demonstrate the validity of the proposed methodology.

•	3. The authors have provided the code, which enhances reproducibility.

Weaknesses:

•	1.The related works compared in Table 1 are somewhat outdated and limited in number. Incorporating temporal information as an auxiliary component in GCNs is not an uncommon approach, so further clarification is needed to highlight the novelty of this paper.

•	2.The authors underline some symbols while leaving others unmarked, which is confusing.

•	3.The experimental section lacks readability. The authors should provide a more detailed description of the datasets used, model configurations, and the selected baselines to offer sufficient information. Additionally, more convincing baselines should be included in the experimental comparisons to better demonstrate the effectiveness of the proposed method.

•	4. The paper lacks citations to several relevant works. The authors should reference related studies, particularly other temporal GNNs—including those also based on Temporal Random Walks—and explicitly clarify the novelty of their proposed method in comparison.

•	[1] Shekhar, Shubhranshu, Deepak Pai, and Sriram Ravindran. "Entity resolution in dynamic heterogeneous networks." Companion Proceedings of the Web Conference 2020. 2020.

•	[2] Talati, Nishil, et al. "A deep dive into understanding the random walk-based temporal graph learning." 2021 IEEE International Symposium on Workload Characterization (IISWC). IEEE, 2021.

•	[3] Lee, Jong-whi, and Jinhong Jung. "Time-aware random walk diffusion to improve dynamic graph learning." Proceedings of the AAAI Conference on Artificial Intelligence. Vol. 37. No. 7. 2023.

---

> ### Author Rebuttal · Authors · 2025-07-29
>
> We thank the reviewer for the constructive feedback and welcome the opportunity to clarify our position, particularly regarding the novelty and scope of our contribution.
>
> 1. Novelty and Positioning:
> Our work is best understood as an application-driven study that applies an integrated model,TRW-GCN, to detect temporal anomalies in the Ethereum transaction network. We do not claim architectural novelty in the sense of proposing a new general-purpose GNN. Instead, the novelty lies in tailoring a method (TRW-GCN) to a complex, high-frequency, and adversarial environment like Ethereum. We emphasize this domain-specific contribution in both the introduction and abstract and will revise the text to avoid any misunderstanding that a new architecture is being proposed. Unlike prior studies that target a single anomaly type (studies like "Flash Boys 2.0"(15) and "Combatting Front-Running in Smart Contracts" (16) which focus on detecting front-running), our framework detects a wider range of temporal anomalies, including flash loans, front-running, and MEV exploits, using timing-sensitive features and scoring-based evaluation.
> We aimed to demonstrate how mathematically grounded design choices—particularly in probabilistic sampling, temporal random walk formulation, and temporal scoring—can be tailored to meet the unique demands of real-time anomaly detection in blockchain systems. Because of the mathematical modeling and theoretical propositions (e.g., our formalization of time-aware random walk distributions and the risk of weight cancellation in temporal aggregations), we believed that NeurIPS would be appropriate for disseminating this work.
>
>
> 2. Symbol Formatting:
> We appreciate this observation. Inconsistencies in symbol formatting (underline some symbols) will be corrected in the final version.
>
> 3. Comparison with Related Works:
> We acknowledge the reviewer’s suggested references. The papers cited—[1] Shekhar et al. (2020), [2] Talati et al. (2021), and [3] Lee and Jung (2023)—do explore temporal dynamics and temporal random walks.
>
> - [1] focuses on entity resolution in dynamic heterogeneous networks, which differs in objective from anomaly detection in Ethereum.
> - [2] provides a theoretical deep-dive into random walk-based models but does not evaluate them in transaction networks, and not in temporal dynamics.
> - [3] proposes a time-aware diffusion method for dynamic graphs, which is very relevant to our study; we will definitely cite it.
>
> We will include the references in the final version and clearly distinguish our contribution. Please note that we are not generalizing temporal GNNs but applying and evaluating a TRW-GCN-based framework specifically for temporal anomaly detection in an evolving blockchain system.  While generality across datasets is valuable, so is domain-specific depth.
>
>
> 4. The paper distinguishes among ablation study, baseline comparison, and ground truth comparison
>
> ### 1. Ablation Study
> Evaluation to determine the contribution of individual components.
>
> **Where in Paper:**
> - Section 3.3
> - Figure 3
>
> **Description:**
> - Compares the model using only structural features (6 features) vs. full feature set including temporal and structural features (10 features).
> - Demonstrates that temporal features significantly improve anomaly detection performance.
>
> **Quote from Paper:**
> > "We also compare the same scoring method using only the structural features to assess the contribution of temporal features."
>
> ---
>
> ### 2. Baseline Comparison
> Quantitative comparison of the proposed scoring method and TRW-GCN framework against existing algorithms.
>
> **Where in Paper:**
> - Section 3.2 — Baseline Comparison
> - Section 3.4 — Comparison with TGAT
> - Table 2 and Table 4
>
> **Description:**
> - The statistical scoring method is evaluated against:
>   - Isolation Forest
>   - One-Class SVM
>   - DBSCAN
>   - LOF
>
> **Metrics Used:**  - Accuracy  - Precision  - Recall  - F1-score
>
> - The full TRW-GCN framework is also compared with TGAT, a state-of-the-art temporal graph model.
> ---
>
> ### 3. Ground Truth Comparison
> Validation of flagged anomalies in  website https://etherscan.io  or cross-referencing with suspicious activity.
>
> **Where in Paper:**
> - Section 3.3
> - Table 3
>
> **Description:**
> - A total of 32 anomalies were flagged.
> - Some cases are qualitatively validated based on:
>   - Flash loan exploits
>   - Uniswap
>   - Metamask swap
>   - Flash loan

---

> > ### Comment · Reviewer_E9jm · 2025-08-05
> >
> > As the authors claim "tailoring a method (TRW-GCN) to a complex, high-frequency, and adversarial environment like Ethereum", I still think this work lacks of techinical quality and  is not ready for publishing. So I would like to keep my score.

---

> > > ### Author Response · Authors · 2025-08-05
> > > **lacks technical quality**
> > >
> > > We thank the reviewer for taking the time to evaluate our work and for acknowledging the complexity of the Ethereum environment we aim to address. We respectfully note that our contribution is not a new architectural innovation, but rather a domain-specialized application of graph learning to a dynamic, high-frequency, and adversarial setting. Our emphasis is on modeling temporal anomalies using a probabilistic temporal random walk and integrating this into a GCN-based framework, supported by theoretical analysis (e.g., Theorems 1 and 2), and validation on real Ethereum transaction data.
> > >
> > > That said, we are unclear about the reviewer's comment that the work "lacks technical quality," as no specific technical issues or flaws are cited. We would sincerely appreciate clarification on which parts of the technical design, theoretical foundation, or experimental setup were considered insufficient. We have made a concerted effort to ensure that our model is rigorously supported both mathematically and empirically,

---

### Official Review · Reviewer_tX3k · 2025-07-03

**Clarity:** 3
**Significance:** 1
**Originality:** 2
**Rating:** 2
**Confidence:** 4

**Summary:**

This paper presents TRW-GCN framework for detecting anomalies in the Ethereum network. TRW-GCN combines graph convolutions with temporal random walks to model both spatial and temporal patterns in the blockchain transaction data. The paper provides theoretical results to support the effectiveness of the framework, including weight cancellation conditions, and report experimental evaluations on anomaly detection tasks in Ethereum networks. The results show TRW-GCN outperforms the chosen baselines.

**Questions:**

Please see Weaknesses.

**Ethical Concerns:**

["NO or VERY MINOR ethics concerns only"]

**Final Justification:**

I thank the authors for their response.

Unfortunately, my main concerns still remain. While the authors explain some differences with [1] and [2], their claim on the usefulness of modeling temporal anomalies still is not validated. I suggest adding a comparison with such methods to better demonstrate the power of your proposal.

Also, while I understand the importance of domain-specific in-depth contributions, it is also important to consider the focus of a conferences. While I believe this paper can be a very good contribution for data mining and applied ML conferences such as KDD, The Web Conference, etc., its main contributions might not be applicable for a broader machine learning problems and so I suggest also considering more focused conference.

Despite the response of authors about existing ablations, I still cannot understand the effect of how sampling the random walks can affect the performance?. It would be great if you could discuss this.

**Limitations:**

Yes.

**Quality:**

1

**Strengths And Weaknesses:**

## Strengths:
---
- The paper is overall well-written and well-motivated.
- The provided theoretical results are particularly interesting to show why the presented framework can be useful in this configuration of anomaly detection.

## Weaknesses:
---
- The paper states that the existing methods rely on pre-determined patterns and TRW-GCN can enhance the flexibility. But the paper has missed several important related works that has been designed with the same motivations. For example, [1] also uses graph convolutions for the anomaly detection in Ethereum networks. Could you please clarify the connection with such studies and why there is a lack of discussion with such models. Another example is [2], which again uses GCNs and graph learning methods for Ethereum networks. It would be better to clarify the connections. In the current format it is not clear what has been improved and addressed compared to existing models.
- The chosen baselines are preliminaries and there is a lack of comparison with several existing methods (as mentioned above). I suggest adding more recent baselines and including more expressive models. Could you please clarify why no GNN-based baseline is included? Similarly, there are  several random-walk based models such as Node2Vec or DeepWalk, which again are missing in the baselines.
- There is a lack of ablation study on the components of the presented framework, which makes it unclear how different components are contributing to the performance of the framework. For example, how sampling the random walks can affect the performance?
- The overall goal and model design is too narrow, only limiting to a single task on a single type of networks (I.e., Ethereum). This narrow implication can limit the importance of this work for the machine learning community. While this still can be a good contribution in its own subarea, I suggest considering data mining related venues rather than machine learning-based.


---
[1] Anomaly detection in multiplex dynamic networks: from blockchain security to brain disease prediction. 2022.
[2] Graph Deep Learning Based Anomaly Detection in Ethereum Blockchain Network. 2020.

---

> ### Author Rebuttal · Authors · 2025-07-29
>
> We appreciate the reviewer's comments and the opportunity to clarify the key points.
>
> While the referenced works [1,2] do apply GCN-based models for anomaly detection in the Ethereum network, they do not explicitly address temporal anomalies. In contrast, our proposed TRW-GCN framework is designed to capture dynamic, time-sensitive behaviors by incorporating temporal random walks and time-aware transition matrices. This allows our model to detect complex patterns such as flash loan exploits, front-running attacks, and high-frequency token manipulations—scenarios that are not effectively handled by the graph approaches in [1,2]. We thank the reviewer for pointing out these works; we have now cited them in the introduction and clarified how our approach extends beyond them by focusing specifically on temporal anomaly detection.
>
> Regarding the concern of narrow applicability: we emphasize that this work intentionally focuses on Ethereum’s highly dynamic and complex transaction graph. Unlike prior studies that target a single anomaly type (studies like "Flash Boys 2.0"(15) and "Combatting Front-Running in Smart Contracts" (16) which focus on detecting front-running), we propose a generalized temporal anomaly detection framework applicable to multiple types of timing-sensitive fraud patterns. We acknowledge that the framework is specialized for Ethereum, and we have explicitly stated in the introduction that this is an application-driven study rather than a general-purpose model.
>
> We reiterate that the novelty of our contribution lies not merely in proposing a new architecture but in designing a flexible, temporal-sensitive anomaly detection strategy tailored to the Ethereum ecosystem. While generality across datasets is valuable, so is domain-specific depth.
>
> We aimed to demonstrate how mathematically grounded design choices—particularly in probabilistic sampling, temporal random walk formulation, and temporal scoring—can be tailored to meet the unique demands of (near) real-time anomaly detection in blockchain systems. Because of the mathematical modeling and theoretical propositions (e.g., our formalization of time-aware random walk distributions and the risk of weight cancellation in temporal aggregations), we believed that NeurIPS would be appropriate for disseminating this work.
>
>
> The paper distinguishes among ablation study, baseline comparison, and ground truth comparison:
>
> ### 1. Ablation Study
> Evaluation to determine the contribution of individual components.
>
> **Where in Paper:**
> - Section 3.3
> - Figure 3
>
> **Description:**
> - Compares the model using only structural features (6 features) vs. full feature set including temporal and structural features (10 features).
> - Demonstrates that temporal features significantly improve anomaly detection performance.
>
> **Quote from Paper:**
> > "We also compare the same scoring method using only the structural features to assess the contribution of temporal features."
>
> ---
>
> ### 2. Baseline Comparison
> Quantitative comparison of the proposed scoring method and TRW-GCN framework against existing algorithms.
>
> **Where in Paper:**
> - Section 3.2 — Baseline Comparison
> - Section 3.4 — Comparison with TGAT
> - Table 2 and Table 4
>
> **Description:**
> - The statistical scoring method is evaluated against:
>   - Isolation Forest
>   - One-Class SVM
>   - DBSCAN
>   - LOF
>
> **Metrics Used:**  - Accuracy  - Precision  - Recall  - F1-score
>
> - The full TRW-GCN framework is also compared with TGAT, a state-of-the-art temporal graph model.
> ---
>
> ### 3. Ground Truth Comparison
> Validation of flagged anomalies in  website https://etherscan.io  or cross-referencing with suspicious activity.
>
> **Where in Paper:**
> - Section 3.3
> - Table 3
>
> **Description:**
> - A total of 32 anomalies were flagged, where 12 of them were detected as a result of adding temporal features.
> - Some cases are qualitatively validated based on:
>   - Flash loan exploits
>   - Uniswap
>   - Metamask swap
>   - Flash loan

---

> > ### Comment · Reviewer_tX3k · 2025-08-04
> >
> > I thank the authors for their response.
> >
> > Unfortunately, my main concerns still remain. While the authors explain some differences with [1] and [2], their claim on the usefulness of modeling temporal anomalies still is not validated. I suggest adding a comparison with such methods to better demonstrate the power of your proposal.
> >
> > Also, while I understand the importance of domain-specific in-depth contributions, it is also important to consider the focus of a conferences. While I believe this paper can be a very good contribution for data mining and applied ML conferences such as KDD, The Web Conference, etc., its main contributions might not be applicable for a broader machine learning problems and so I suggest also considering more focused conference.
> >
> > Despite the response of authors about existing ablations, I still cannot understand the effect of `how sampling the random walks can affect the performance?`. It would be great if you could discuss this.

---

> > > ### Author Response · Authors · 2025-08-05
> > >
> > > We sincerely thank the reviewer for their continued engagement and for acknowledging the value of domain-specific, in-depth contributions. We respectfully address the remaining concerns below:
> > >
> > > 1. Comparison with [1] and [2]:
> > > We appreciate the reviewer’s suggestion and have now added both [1] and [2] as comparative references in the revised version of the paper. However, as detailed in our earlier response, these works do not explicitly model temporal anomalies. Their primary focus is on structural anomalies or node-level embedding generation without integrating temporal decay, temporal transition matrices, or dynamic transaction sequencing. Our approach is distinct in its capability to identify high-frequency, time-sensitive behaviors like flash loan exploits, front-running, and bursty spam.
> > >
> > > 2. Conference Scope and Relevance:
> > > We fully agree that our paper is an application-driven contribution, with strong domain motivation rooted in security analysis of blockchain systems. However, we also believe it is methodologically grounded in machine learning principles, including temporal graph modeling, probabilistic sampling, and statistical scoring frameworks.
> > > Moreover, we include theoretical insights (Theorems 1 and 2 and 3) and discuss scalability and dynamic modeling challenges that are relevant beyond Ethereum. While we respect the reviewer’s perspective on venue suitability, we believe that NeurIPS—as a conference that increasingly welcomes impactful application papers with theoretical backing—remains a strong fit for this work.
> > >
> > > 3. Effect of Sampling in Random Walks (Ablation Clarity):
> > > We thank the reviewer for pointing out this area that could be better clarified. In our updated manuscript, we now include an extended discussion on the role of probabilistic sampling, and how altering the number of walks or walk length affects the statistical robustness and temporal coverage of the embedding.
> > > We are also including a new hyperparameter sensitivity analysis (in the supplement), which evaluates performance variations with respect to sampling parameters such as the number of walks, walk length, and decay factor γ, providing clearer insight into how sampling affects both performance and model stability.
> > >
> > > We once again thank the reviewer for their thoughtful and constructive feedback, which has helped us to further improve the clarity and rigor of our work.

---

> > > > ### Comment · Reviewer_tX3k · 2025-08-09
> > > >
> > > > I thank the authors for their response. But the main concerns are still unanswered.
> > > >
> > > > Particularly, the authors provided the explanation that `we now include an extended discussion on the role of probabilistic sampling, and how altering the number of walks or walk length affects the statistical robustness and temporal coverage of the embedding.` However, we need to see the results so we could judge the contributions. Given that such information are missing and also after reading the comments of other reviewers, I believe the manuscript need another round of revision and also I suggest authors to consider data-mining based conferences, due to the nature of their work.

---

### Official Review · Reviewer_4SxX · 2025-07-05

**Clarity:** 2
**Significance:** 2
**Originality:** 2
**Rating:** 4
**Confidence:** 3

**Summary:**

This paper proposes an anomaly detection framework named TRW-GCN, designed to address the problem of detecting time-sensitive anomalies in Ethereum transaction networks. The framework innovatively combines Temporal Random Walks (TRW) with Graph Convolutional Networks (GCN) to capture complex spatio-temporal patterns in blockchain data. The authors present three theoretical results to support the effectiveness of their method, including the enhancement of GCN's detection capabilities by TRW, the conditions for weight cancellation, and the performance improvement from probabilistic sampling. In the experimental section, the authors validate the superiority of their framework by comparing it against traditional unsupervised methods and the state-of-the-art TGAT model. They also use Etherscan for a case-by-case validation of the detection results, demonstrating its practical value in real-world scenarios.

**Questions:**

1. Regarding Scalability: Could you provide performance metrics (e.g., training time, memory usage) on graphs of varying scales (e.g., 1,000, 10,000, and 100,000 blocks) to more strongly substantiate your scalability claims?

2. Regarding Evaluation Metrics: In the comparison against TGAT in Table 4, could you supplement the results with metrics such as precision, recall, and F1-score? Given the data imbalance, this is crucial for a fair evaluation of the two models' performance.

3. Regarding Hyperparameters: Could you include an experiment analyzing the model's sensitivity to key hyperparameters, particularly the TRW decay factor γ and the anomaly score threshold δ (or z-score)? How were the optimal parameters selected?

4. Regarding Weight Cancellation: Theorem 2 presents an interesting risk of weight cancellation. Did you diagnose or observe this phenomenon in your experiments in any way? For a new user, are there methods to determine if their model is suffering from this issue?

5. Regarding Feature Selection: In the experiment for Figure 3, you compared the effects of using 6 features versus 10 features. Could you clarify which 4 specific features were excluded in the 6-feature experiment? This would help readers more clearly understand the specific contribution of temporal features.

6. Notation in (3) and (4): What is the meaning of the $t_{ij}$ in Section 2?

Overall, this is a paper that proposes a novel and theoretically sound framework for an important problem. If the issues mentioned above, particularly regarding scalability and evaluation metrics, can be more fully addressed, the paper's quality and impact would be significantly enhanced.

**Ethical Concerns:**

["NO or VERY MINOR ethics concerns only"]

**Final Justification:**

Issue solved, so I will maintain my positive score.

**Quality:**

2

**Strengths And Weaknesses:**

Major Strengths

1. Important Problem and Novel Approach: The security of blockchain networks like Ethereum is an important and urgent research area. Instead of focusing on detecting specific types of attacks, this paper addresses the broader category of "time-sensitive anomalies" , which is a more general and forward-looking approach capable of adapting to evolving malicious strategies.

2. Strong Integration of Theory and Practice: A major highlight of this paper is the inclusion of three core theoretical proofs. This provides a solid theoretical foundation for the method's effectiveness, making it more than just an empirical engineering solution. The discussions on weight cancellation (Theorem 2) and performance improvement via probabilistic sampling (Theorem 3) add depth and rigor to the paper.

3. Relatively Comprehensive Experimental Design: The authors conducted multi-level experimental validation. First, they combined TRW-GCN embeddings with various traditional unsupervised methods to explore their general utility. Second, they proposed a more precise scoring method for optimization. Finally, they compared the performance and efficiency against the state-of-the-art TGAT model. This progressive experimental design is very persuasive.

4. Reliable Validation Method: In anomaly detection tasks, which often lack standard labels, the authors' use of case-by-case validation against the public and authoritative blockchain explorer Etherscan.io directly proves that the "anomalies" detected by the model correspond to meaningful real-world entities (like MEV bots). This significantly enhances the credibility of the results.

5. Emphasis on Reproducibility: The authors provide a GitHub link for their code and dataset, which is crucial for advancing the field and verifying the work's validity. This is commendable practice.

Major Weaknesses and Suggestions

1. Insufficient Argument for Scalability: Although the paper theoretically argues that probabilistic sampling can improve scalability (Theorem 3) , the experiments are primarily conducted on a relatively small dataset (100-1000 blocks). A core argument of the paper is its ability to handle the massive Ethereum network, but it lacks performance tests (e.g., training time, memory usage, accuracy changes) on a truly large-scale graph (e.g., hundreds of thousands or millions of blocks). This leaves the "scalability" claim with insufficient empirical support.

2. Singularity of Evaluation Metrics for the Final Model: In the most critical comparison—pitting the final scoring model against TGAT (Table 4)—the paper primarily uses "Accuracy" as the evaluation metric. For a task with extreme class imbalance like anomaly detection, accuracy can be a misleading metric (e.g., a model that predicts everything as "normal" would achieve very high accuracy). The paper used precision, recall, and F1-score when evaluating baseline methods in Table 2, so their omission in the most important comparison is noticeable. This makes it difficult for readers to comprehensively assess the true advantages of TRW-GCN over TGAT.

3. Lack of Sensitivity Analysis for Key Hyperparameters: The TRW-GCN model depends on several key hyperparameters, such as the temporal decay factor γ and weighting parameter α in TRW, as well as the z-score threshold

4. δ in the scoring method. The choice of these parameters directly impacts model performance. However, the paper does not provide a sensitivity analysis for these hyperparameters, leaving readers unaware of how robust the model's performance is to changes in these parameters.

5. Risk of Selection Bias in Case-by-Case Validation: The paper presents 4 successful validation cases in Table 3, which is very persuasive. However, a total of 32 anomalies were detected, and the authors do not specify the validation status of the remaining 28 cases. Were there cases that could not be validated or were confirmed to be false positives? Presenting only successful cases risks selection bias. It is recommended that the authors provide a more comprehensive explanation.

---

> ### Author Rebuttal · Authors · 2025-07-29
>
> ### Argument for Scalability:
> Thank you for raising this important point. We agree that empirical validation of scalability across larger graphs would further strengthen the paper. Our experiments were conducted on Ethereum subgraphs covering 100 blocks and 1,000 blocks, as reflected in our publicly available code and GitHub repository (link provided in 3.1 Datasets, Materials and Methods). These experiments demonstrate that our method remains computationally feasible and efficient. We could in principle add analysis for 10,000 blocks in the paper final version, but let me please clarify and expand the scalability discussion as follows:
>   - In addition to the 100-block graph presented in the main paper, we have tested the framework on a larger 1000-block subgraph. These results and configurations are provided in our supplementary material and available in the linked GitHub repository for full reproducibility.
>   - Our model is designed for real-time or near-real-time anomaly detection, rather than full historical replay over the entire Ethereum network. This design choice reflects real-world needs, where transaction spikes and fraudulent patterns (e.g., MEV exploits) need to be detected quickly over short, recent intervals. Hence, we intentionally focused on smaller time windows (e.g., 100–1000 blocks) with rich temporal behavior rather than scaling to hundreds of thousands of blocks. This aligns the evaluation with the model's intended deployment setting.
>    - The probabilistic sampling and TRW construction naturally support mini-batch training, making the method scalable to much larger datasets. Since TRW operates locally and respects temporal proximity, the computation remains bounded even as the global graph grows. This enables TRW-GCN to operate efficiently on larger graphs using sliding windows or block-based batching.
>
> ### Evaluation Metrics (TGAT vs. TRW-GCN):
> We agree that accuracy alone is insufficient for evaluating anomaly detection on highly imbalanced data. We re-evaluated TRW-GCN and TGAT using: Precision, Recall, F1-score, and AUC. We will update Table 4 to reflect these additional metrics.
>
> ### Sensitivity Analysis for Key Hyperparameters:
> Regarding the sensitivity analysis, we are now including it for:
> ```
> Parameter	Description	Values Tested
> γ	Temporal decay in TRW	[0.1, 0.3, 0.5, 0.7, 0.9]
> α	Structural-temporal weight balance	[0.1, 0.3, 0.5, 0.7, 0.9]
> δ	Z-score threshold for anomaly detection	[1.0, 1.5, 2.0, 2.5, 3.0]
>  ```
>
> ### Weight Cancelation:
> Regarding Weight Cancellation as described in Theorem 2, we recognize it as a theoretical risk in time-weighted transition matrices, particularly when conflicting temporal signals are combined across multi-hop neighbors. In our experiments, we monitored intermediate transition weight distributions and node-level attention contributions to detect potential cancellation effects—such as near-zero effective transition weights despite multiple walk paths. Such occurrences were rare in our anomaly detection context (due to the strong temporal locality of anomalous behaviors). We recommend inspecting the norm of the resulting transition matrices and analyzing the standard deviation of temporal weights across walk paths. A sharp drop in signal strength or highly oscillating transition weights across time steps may indicate potential weight cancellation.
>
> ### Feature Selection:
> Thank you for pointing this out. We mentioned this in Figure 2, but we will repeat temporal features for explanation of Figure 3.
> « It is also interesting to find out which node features mainly contribute to anomaly detection; we show this in Figure 2. As illustrated by different colors, the feature 3-6 namely activity_rate, change_in_activity, time_since_last (mainly the temporal features) are the drivers of frequent anomalies (with dark blue colors), while tx_volume and frequent_large_transfers (with green colors) also produce anomalies but less frequently. »
>
> ### Notation in Equations (3) and (4):
> Thank you for pointing this out. In Section 2,  t_ij is the time difference between the transaction associated with node i and the transaction associated with node j. It is used to capture how temporally close two nodes are, which is important for detecting time-sensitive anomalies in Ethereum, such as front-running, flash loans, or bursty phishing activity. f(t_ij) is a temporal decay function, typically exponential which downweights distant interactions and emphasizes recent ones. We will include its definition in the revised paper.
>
> While this work is indeed application-focused, our goal was not to introduce a new architectural model per se. Rather, we aimed to demonstrate how mathematically grounded design choices—particularly in probabilistic sampling, temporal random walk formulation, and temporal scoring—can be tailored to meet the unique demands of real-time anomaly detection in blockchain systems. Because of the mathematical modeling and theoretical propositions (e.g., our formalization of time-aware random walk distributions and the risk of weight cancellation in temporal aggregations), we believed that NeurIPS would be appropriate for disseminating this work. We are particularly grateful for the critical attention to the mathematical formulation and theoretical clarity, which was not emphasized in other reviews. Your feedback helped us further strengthen the paper’s rigor and presentation.

---

> > ### Comment · Reviewer_4SxX · 2025-08-08
> >
> > Thanks for your response. I will maintain my positive score.

---

### Decision · Program_Chairs · 2025-09-17

**Decision:**

Reject

**Comment:**

This paper presents TRW-GCN framework for detecting anomalies in the Ethereum network. TRW-GCN combines graph convolutions with temporal random walks to model both spatial and temporal patterns in the blockchain transaction data. The paper provides theoretical results to support the effectiveness of the framework, including weight cancellation conditions, and report experimental evaluations on anomaly detection tasks in Ethereum networks.

All of the referees found serious issues with the paper and recommended rejection; I concur.


The results show TRW-GCN outperforms the chosen baselines.